# Neutrophilic Asthma—From Mechanisms to New Perspectives of Therapy

**DOI:** 10.3390/jcm14207137

**Published:** 2025-10-10

**Authors:** Ilona Iwaszko, Krzysztof Specjalski, Marta Chełmińska, Marek Niedoszytko

**Affiliations:** Department of Allergology, Medical University of Gdansk, 80-210 Gdansk, Poland; ilonaiwaszko@gumed.edu.pl (I.I.); allergy@gumed.edu.pl (M.C.); mnied@gumed.edu.pl (M.N.)

**Keywords:** non-eosinophilic asthma, neutrophilic asthma, neutrophils

## Abstract

Neutrophilic asthma (NA) is an inflammatory phenotype of asthma, characterized by predominantly neutrophilic infiltrations in bronchial mucosa. It is usually diagnosed on the basis of high neutrophil count in induced sputum (from >40% to >76%) with low eosinophils (<2%). The prevalence of NA ranges from 16% to 28% of the adult asthma population depending on the definitions and study methods applied. A clinical picture of NA is characterized by late onset of symptoms, higher exacerbation rate, lower level of symptoms control, and poorer response to steroids compared to eosinophilic phenotype. Comorbidities such as obesity and GERD as well as the influence of environmental factors (air pollution, smoking, bacterial infections) contribute to the development and severe course of the disease. NA is T2-low disease with predominantly Th1/Th17-type inflammation. Neutrophils are key cells responsible for initiating and sustaining inflammation. In addition to their primary functions like phagocytosis, degranulation, and NETosis, neutrophils release several pro-inflammatory cytokines (IL-1α, IL-1β, IL-6, TNF) and chemokines (CXCL-1, -2, -8, -9, -10) responsible for the recruitment of other neutrophils or T cells. Increasing knowledge about the biology of neutrophiles and their role in asthma results in new potential therapies that could improve control of NA, particularly new biologicals targeting Th1/Th17-related cytokines. In this review, we discuss the prevalence, mechanisms, and clinical features of neutrophilic asthma. Furthermore, current therapeutic options and some promising perspectives for the near future are presented.

## 1. Introduction

Asthma is characterized by chronic airway inflammation leading to bronchial hyperreactivity and, as a consequence, recurrent episodes of wheezing, shortness of breath, chest tightness, and cough which vary in time and intensity [1]. Asthma is a heterogeneous disease as patients vary in terms of inflammation patterns, concomitant diseases, exacerbating factors, as well as response to therapy. Eosinophilic asthma (EA) is the most common inflammatory phenotype. Chronic T2-high inflammation is characterized by overexpression of pro-inflammatory cytokines (IL-4, IL-5, IL-13) and eosinophilic infiltrations in bronchial mucosa [2]. Clinical characteristics of EA include early onset of symptoms, high prevalence of rhinitis/sinusitis, as well as a significant influence of airborne allergens or non-steroidal anti-inflammatory drugs (NSAIDs). However, patients with this phenotype usually have good response to ICS treatment [1]. Moreover, in recent years they have benefited from modern biological therapies targeting IgE, IL-5, IL-5R, or IL-4/-13R.

Contrary to EA, phenotypes related to T2-low inflammation (neutrophilic and paucigranulocytic asthma) are not well defined. Moreover, in clinical practice, no biomarkers are routinely used to diagnose them, assess severity of symptoms, or predict prognosis. In clinics, they are characterized by later onset, coexisting obesity, tobacco smoking, as well as a worse response to ICS compared to EA [3]. So far no biological drugs targeting neutrophils have been registered. As a result, patients suffering from T2-low asthma achieve control of symptoms less often and have more exacerbations compared to EA counterparts.

Considering the significant progress in the therapy of EA that followed investigating its mechanisms, it can be speculated that better understanding of non-EA would also lead to defining new drug targets and biomarkers. Consecutively, personalization of management would ameliorate its efficacy and safety.

This review discusses prevalence, mechanism, and clinical features of neutrophilic asthma. Furthermore, current therapeutic options and new perspectives for the future are presented.

## 2. Definition and Prevalence of Neutrophilic Asthma

### 2.1. Diagnosing Neutrophilic Asthma

Cross-sectional cluster analyses have identified four major asthma subgroups based on the sputum cellular patterns: eosinophilic, neutrophilic, mixed, and paucigranulocytic. Eosinophilic phenotype is the best defined one with proposed thresholds of blood eosinophilia (>0.3 × 10^9^/L or >0.15 × 10^9^/L); sputum eosinophilia ≥ 2% [1]. In contrast, neutrophilic asthma has not been well defined yet. Cluster analyses demonstrate that clinical features distinct from eosinophilic asthma are present in study groups characterized by a high neutrophil count in induced sputum (from >40% to >76%) with low eosinophils (<2%) [4,5,6,7].

There is no consensus on a specific cut-off point for neutrophilic asthma within BAL. However, the findings of certain studies suggest that values of neutrophil rate ≥5% may prove to be useful in the identification of a neutrophil-predominant phenotype, especially for children [8,9].

As far as neutrophil blood count is concerned, a cut-off of 5.0 × 10^9^/L has been used in several studies [10]. Contrary to sputum or BAL neutrophilia, it is not associated with clear clinical phenotypes so it probably does not define neutrophilic asthma well.

So far there are no recommended, easily available and reliable biomarkers that could facilitate the diagnosing of NA.

In conclusion, diagnosing neutrophilic asthma is challenging. There are no tests that can be easily implemented in everyday practice. Cut-offs for induced sputum neutrophil count vary significantly which may stem from heterogeneity of the population with NA.

Detailed data regarding proposed cut-offs for diagnosing neutrophils-predominant asthma are shown in Table 1.

### 2.2. Estimated Prevalence of Neutrophilic Asthma

As there are no explicit criteria based on cut-offs for cell concentration in BAL or sputum, the prevalence of neutrophilic asthma (NA) mostly ranges from 16% to 28% of the adult asthma population depending on the definitions and study methods applied [7,17,18]. However, some studies demonstrate much lower rates (4–5%), while other studies found neutrophilic asthma to be the predominant phenotype (57%) [17,19,20]. In a large study of 995 patients tested repeatedly over time, 47% of patients did not have eosinophilia at any time and another 20% only sporadically suggesting a significant role of non-eosinophilic pathways in majority of the cases [21].

Among infants and children (6 months–3 years of age) with recurrent wheezing, neutrophilia was found in 65% sputum samples. Interestingly, neutrophils prevailed in infants with mild-to-moderate course of the disease while severe patients had predominantly eosinophilic inflammation [22]. Among 2800 children with poorly controlled asthma, 16% had neutrophilic phenotype confirmed by means of BAL [23].

### 2.3. Is Neutrophilic Asthma a Phenotype or a Lab Artifact?

The prevalence of neutrophilic asthma is diverse partly due to the fact that cellular patterns of inflammation vary over time. It has been demonstrated that patients with EA experience exacerbations [24]. Moreover, sputum eosinophil and neutrophil counts seem to be significantly affected by the therapy of asthma. Cowan et al. compared induced sputum of 94 patients with asthma in two time points: steroid-naïve and upon treatment. In the former, 67% of patients were eosinophilic, 31% paucigranulocytic and 2% mixed; there were no neutrophilic subjects. After the introduction of ICS, 39% were eosinophilic, 46% paucigranulocytic, 3% mixed, and 5% neutrophilic. Sputum neutrophils rate increased upon treatment from 19.3% to 27.7% [25]. In another study, neutrophils were found in bronchial biopsies in patients with typical clinical pattern of EA with atopy and elevated serum IgE [14].

Moreover, blood neutrophilia is affected by several concomitant conditions. It is well known that neutrophilia accompanies hematologic and non-hematologic malignancies, chronic inflammatory diseases (e.g., colitis ulcerosa, Crohn’s disease, hepatitis, rheumatoid arthritis, etc.), thrombosis, asplenia, and several endocrinopathies. In some cases, it is induced by therapy with beta-blockers, lithium, or systemic glucocorticosteroids, including, often forgotten by patients, joint injections of slow-release formulas [26]. It has been demonstrated that smokers have higher neutrophilia compared to non-smokers [27].

The variability of airway neutrophilia, concomitant influence of eosinophilic inflammation, and possible effects of coexisting pathologies impede the interpretation of several studies in this field. As a consequence, there is no generally accepted diagnostic criterion for neutrophilic asthma. Moreover, its very existence is sometimes questioned. Some researchers believe that this is COPD rather than asthma. Others dismiss airway neutrophilia as a marker of corticosteroid exposure or tobacco smoking without causative relation to asthma itself [28]. In our view, diagnosis should not be based solely on neutrophils level. Instead, clinical characteristics of the patient should be thoroughly analyzed.

Furthermore, in everyday practice as well as in a large body of the literature, the division into eosinophilic (i.e., mostly T2-high) and non-eosinophilic (i.e., T2-low) asthma is often used. In this approach, non-eosinophilic asthma includes neutrophilic and paucigranulocytic phenotypes. From the practical point of view, such simplification is quite useful as it allows for a determination of a major endotype in a given patient and allocation of proper biological therapy. However, it often leads to using the terms ‘non-eosinophilic’ and ‘neutrophilic’ asthma interchangeably. As a result, it is difficult to ascertain how many non-eosinophilic cases had primary paucigranulocytic phenotype and how big the proportion of predominantly neutrophilic asthma truly is in this entity. Some authors have demonstrated that paucigranulocytic asthma is twice as common as neutrophilic (38% vs. 16%) [7].

## 3. Clinical Characteristics of Neutrophilic Asthma

One of the earliest studies linking neutrophilic inflammation with distinct clinical features of asthma was a report from the Mayo clinic presenting cases of fatal asthma attacks. The fatalities had prevailing neutrophilic infiltrations in the airway submucosa in contrast to eosinophilic inflammation usually found in patients who had not had acute events [29].

### 3.1. Asthma Onset

Cluster analyses have demonstrated that groups with predominantly neutrophilic asthma are relatively older and most of the participants were diagnosed with asthma in adulthood [5,17]. In the study by Moore, half of the group was diagnosed with asthma at the age > 12 years [6]. Another study demonstrates that the median age at diagnosis was 56 years [4]. In most of the studies, women prevailed over men [4].

Studies on asthma in elderly have shown that the proportion of patients with sputum neutrophilia is increasing with age [30]. In contrast, an allergic component is also found to be less frequent with advanced age [30]. On the other hand, neutrophilic profiles are often found in patients reporting allergen-induced exacerbations over time or individuals with elevated total and specific IgE [11].

### 3.2. Clinical Course of the Disease

Cluster analyses have demonstrated that neutrophilic inflammation is associated with poorer control of asthma with higher incidence of daily symptoms and awakenings as well as higher usage of rescue treatment [17,31,32]. Unsurprisingly, neutrophilic asthma is related to lower disease-related quality of life [17].

Sputum and bronchial neutrophils as well as the enhanced expression of Th17-related cytokines are strongly associated with the severity of asthma [33,34]. Moore et al. have demonstrated that 38% of NA patients have severe asthma [6]. In the Chinese population, although neutrophilic asthma was found to be uncommon (4.3% of participants), 60% of its cases were assessed to be severe [17].

Population with neutrophilic asthma is also characterized by a higher exacerbation rate. In a large Dannish study analyzing the course of asthma over 8 years, blood neutrophilia > 4.85 G/l doubled the risk of moderate exacerbations of asthma, defined as those requiring OCS intake [35]. Higher exacerbation rate was also found to be related to the expression of cytokines associated with neutrophilic inflammation [33]. Bullone et al. have shown that 17% of the NA group had more than one exacerbation per year compared to 9% in non-neutrophilic cases [11]. In another study, the NA group had the highest rate of emergency department visits as well as hospitalizations due to asthma attacks [6]. In contrast, in a study by Welsh et al. based on repeated induced sputum assessment, persistent neutrophilia was not associated with number of exacerbations or shorter time to first exacerbation [36]. Inconsistencies between the studies may result from variability of airway neutrophilia and heterogeneity of the population with NA.

Neutrophilic phenotype is characterized by a poorer response to first-line asthma treatment—ICS [4]. Some authors claim that neutrophilic inflammation is responsible for about 50% of corticosteroid resistant/insensitive asthma cases [37]. Usually, to achieve comparable clinical outcomes, higher dose of ICS should be used than in EA. In cluster analyses, NA patients are treated, on average, with higher doses of ICS, and up to 40% require high doses of ICS [6]. Some studies show that the dose of ICS is about twice as high compared to that for other phenotypes [11].

The severity of asthma and relatively frequent exacerbations lead to higher use of systemic steroids either continuously or upon exacerbations [6]. The prevention of exacerbations with biological drugs is not as available as that for the more common EA. This is due to the fact that the majority of currently used biological drugs are indicated for the treatment of T2-high inflammation, so patients with the neutrophilic phenotype do not benefit from them [1]. Again, elevated blood neutrophil count was associated with uncontrolled asthma in patients receiving biologics [32].

### 3.3. Pulmonary Function

Patients with neutrophilic asthma have significantly lower FEV_1_% and FEV_1_/VC and smaller reversibility after SABA compared to the eosinophilic group [11]. Substantial proportion of the group has fixed obstruction. Choi et el. found that NA patients constitute majority of the fixed obstruction group with remodeling [38]. A 10-fold increase in neutrophil count was associated with a reduction in postbronchodilator FEV_1_ of 92 mL [39]. Some authors do not confirm that, while others indicate that the neutrophilic group has more severe obstruction compared to the paucigranulocytic, but not the eosinophilic group [4,6,7].

Some authors have demonstrated functional markers of air trapping (decreased FVC%, increased FRC%). The highest numerical value of RV was observed in severe neutrophilic asthma, and this group had the highest proportion of air trappers [11].

### 3.4. Resistance to ICS

Neutrophilic asthma is associated with treatment with higher doses of ICS, more frequent oral steroid use, and more frequent hospital admissions (65% do 28%) [6]. It is estimated that 50% corticosteroid-resistant/insensitive asthma cases are characterized by neutrophilic inflammation [37]. The limited response to such treatment can be explained by several reasons. Firstly, CS has been shown to inhibit the inflammatory pathway characterized by the predominance of T2 cytokines such as IL-4 and IL-5. The consequence of this pathway being inhibited is a reduction in eosinophil production and survival, with a resultant alleviation of inflammation in the airways [40,41]. While ICS promote apoptosis of eosinophils, they have been demonstrated to inhibit apoptosis of neutrophils [42,43]. The cytokines IL-8, IL-17, and TNF-α, which are present in Th1/Th17 inflammation, contribute to neutrophil recruitment and activation in mechanisms not related to ICS. IL-17 induces the expression and activity of transcription factor CEBPB which, in turn, enhances the production of lipocalin-2 (LCN-2) and serum amyloid A (SAA), contributing to resistance to ICS [44,45].

### 3.5. Concomitant Diseases

Patients with neutrophilic asthma are characterized by elevated levels of CRP and fibrinogen, suggesting that systemic inflammation may be hallmark of this phenotype [7,46]. Inflammation may also stem from concomitant diseases. Cluster analyses have demonstrated that NA is often accompanied by obesity and GERD. A substantial proportion of patients are current smokers or ex-smokers [47].

#### 3.5.1. Obesity

Obesity is a chronic disease of complex etiology, defined as abnormal or excessive fat accumulation. Worldwide, 500 million people are obese; this is equivalent to 11% of adult men and 15% of women [48]. Patients with asthma are even more affected, having a prevalence exceeding 20% [49]. Obesity elevates the risk of asthma and obese patients tend to have uncontrolled symptoms, more frequent and severe exacerbations, as well as a worse response to ICS.

In obesity, adipose tissue propagates chronic inflammation both locally and systemically [50]. Fat cells produce several adipokines such as leptin, resistin, lipocain 2, IL-6, TNF, and IL-1β, which recruit and activate cells of the innate immune system and induce systemic inflammation [51,52,53,54]. Leptin has pro-inflammatory properties, improves neutrophils’ function, and increases their recruitment into organs, including the lungs [50,55,56]. This explains correlations that have been shown between blood neutrophil counts and BMI [57,58]. In the female asthmatic population, 1.0% increase in neutrophil blood counts was associated with one unit increase in BMI [59].

In cluster analyses, obesity-related asthma has shown predominantly non-Th2 endotype, with later onset and severe symptoms, more prevalent in females. In a study assessing the severity of adult-onset asthma, obese patients made up 16% of the uncontrolled cases and only 8% of the controlled asthma [60]. In a cohort of 233 subjects with asthma, the highest neutrophil counts were observed in a group with severe obesity. A total of 83.3% of obese subjects had blood neutrophils ≥ 4.0 × 10^9^/L compared to 31.5% in the normal weight subgroup. Compared to the reference group (BMI < 30 kg/m^2^, blood neutrophils < 5 × 10^9^/L), BMI ≥ 30 kg/m^2^ with blood neutrophils ≥ 5 × 10^9/^L was associated with uncontrolled asthma [61].

Weight loss interventions have been demonstrated to be effective in improving both asthma control and lung function [62].

#### 3.5.2. GERD

GERD is a very common comorbidity in asthma. Most of the studies indicate that 25–35% of patients are affected with a much higher incidence in difficult-to-treat and severe asthma [63,64,65]. Simpson et al. have demonstrated that GERD is more frequent in neutrophilic than in eosinophilic asthma [66]. Relations between asthma and GERD are complex as they may interact in a vicious cycle.

First, GERD affects the respiratory system in mechanisms of micro-aspirations (‘reflux theory’) as well as bronchoconstriction related to common vagal innervation of bronchi and esophagus (‘reflex theory’). In healthy individuals, GERD is one of the most common causes of chronic cough and vocal cord dysfunction manifested by hoarseness [67]. GERD-related cough is related to elevated sputum neutrophil counts and overexpression of pro-inflammatory cytokines including IL-8 [35]. These effects are reversible upon treatment with proton pump inhibitors (PPI) [68].

On the other hand, asthma can aggravate GERD by lung hyperinflation and raising the pressure gradient between the chest and the abdomen. Moreover, bronchodilators used in the therapy of asthma lead to the relaxation of lower esophageal sphincter [67]. These effects seem to be particularly important in obese patients with asthma. The triangle of asthma, GERD, and obesity correlates with neutrophilic inflammation in lower airways [69].

#### 3.5.3. Cigarette Smoking

It is estimated that 20–25% of the asthma population have some smoking habits [47]. It has been well documented that exposure to smoke substantially increases risk of asthma in children. Smoking patients with asthma have a higher percentage of neutrophils in induced sputum and a poorer response to treatment with inhaled corticosteroids compared to non-smoking counterparts [7,70]. Combustible cigarette smoking increases levels of several pro-inflammatory cytokines including IL-17a, IL-6, and IL-8. Concomitant exposure of human epithelial cells to IL-17a, smoke, and airborne allergens leads to secretion of IL-6 and IL-8 associated with neutrophilic inflammation [71]. This phenomenon has also been confirmed in an animal model. Response to ovalbumin in sensitized rats was increased by cigarette smoke exposure and led to neutrophilic infiltrations of the pulmonary tissue [72].

Smoking patients with asthma and low FEV_1_ have also been shown to have an increased number of IRF5+ macrophages (associated with Th1 responses) and a reduced number of IL10+ macrophages (with anti-inflammatory effects). It has been suggested that this may be the result of the combined effects of smoking and chronic asthma [73].

Comparison of the most clinical characteristics of EA and NA is presented in Table 2.

NA shares some clinical features with paucigranulocytic asthma. Analysis by Deng et al. demonstrated three distinct clinical clusters of paucigranulocytic asthma. One of them was smoking-associated and characterized by older age, later asthma onset, and increased risk of exacerbations. Another two had some features of EA with rhinitis and mild symptoms [74]. Due to several clinical similarities, non-eosinophilic asthma phenotypes are often perceived as one entity. Some authors propose using more practical, ‘clinically-centered’ phenotypes based on concomitant disease (‘obesity-related asthma’ or ‘smokers’ asthma’) instead of cellular assessment [75].

However, according to most authors, paucigranylocytic asthma represents “benign” phenotype with good response to treatment [76]. Ntontsi et al. demonstrated that only 15% of patients with paucigranulocytic asthma have uncontrolled symptoms. Pulmonary function has been found to be significantly better compared to that of patients with eosinophilic or neutrophilic asthma [19].

## 4. Neutrophils in Asthma

Neutrophils are the most abundant class of white blood cells in human circulation and are a crucial component of the immune system, particularly in antibacterial and antifungal responses [77,78]. Their best-known functions include phagocytosis, degranulation, and release of neutrophil extracellular traps (NETosis) [79,80,81]. Neutrophils secrete several pro-inflammatory cytokines (IL-1α, IL-1β, IL-6, TNF) and chemokines responsible for the recruitment of other neutrophils (CXCL-1, -2, -8) or T cells (CXCL-9, -10) [82,83]. On the other hand, they also regulate tissue repair and remodeling. Recent studies have shown heterogeneity of neutrophils and their functional compartmentalization [84,85,86]. Moreover, neutrophils populations in tissues change their phenotype (N1 or N2-predominant) depending on tissue microenvironment with either prevailing pro-inflammatory activity or promoting of regeneration and growth. Such functional plasticity of neutrophils enables their adaptation to the characteristics of tissue and the current situation.

In the course of asthma, neutrophils migrate into the airways in the early phases of the disease and subsequently release various inflammatory mediators. Their recruitment may be initiated by viral or bacterial infections, air pollution, tobacco smoke, or other irritants. Epithelial injury triggers the activation of epithelial cells and tissue macrophages to produce cytokines and chemokines including IL-1α, Il-1β, IL-6, IL-8, IL-17, and IL-23. IL-8 is believed to be the most potent neutrophil chemoattractant in the lungs, promoting gradient-dependent migration, i.e., from lower to higher concentration of the cytokine [87]. Neutrophils respond to IL-8 signaling via CXCR1/CXCR2 receptors on the cell surface. The binding leads to directed cell movement across the endothelium and epithelium into the airway lumen. Furthermore, the capacity of neutrophils to secrete IL-8 has been demonstrated. Thus, initial stimulation by IL-8 is consecutively enhanced by a feedback loop that promotes neutrophil recruitment to the airways by neutrophils [88].

### 4.1. Direct Effects of Neutrophils on Bronchial Mucosa

Degranulation of neutrophils is one of the main mechanisms stimulating inflammatory bronchial response in patients with asthma. Contents of neutrophilic granules such as NE, MPO, cathepsin G, and proteinase 3 damage the epithelial barrier, destroy cells, and induce the secretion of pro-inflammatory cytokines by epithelial cells (Figure 1) [89,90,91]. MPO, a cationic enzyme located in primary azurophilic granules, has been shown to catalyze ROS production and to be involved in NET formation and stabilization [92,93,94].

Increased production of reactive oxygen species (ROS) in neutrophilic asthma contributes to chronic oxidative stress, which in turn promotes lung tissue remodeling and impairs bronchial epithelial regeneration [95,96,97,98]. The excess of ROS leads to a number of detrimental effects, including lipid peroxidation of cell membranes, denaturation of proteins and enzymes, DNA damage, and activation of inflammatory factors such as NF-κB [99,100,101,102,103].

NETs formed by neutrophils in the context of asthma disrupt the tight junctions of the bronchial epithelium, leading to the efflux of intracellular components [104,105]. NETs can trigger inflammatory responses by damaging airway epithelial cells, inducing IL-8 production and activating eosinophils to release their granules [106]. The protein HMGB1, a component of NET, has been demonstrated to enhance the expression and secretion of pro-inflammatory cytokines, alarmins, and growth factors (e.g., TNF-α, TSLP, MMP-9, and VEGF). These mediators play a pivotal role in the course of neutrophilic asthma [107].

### 4.2. Neutrophils-Associated Cytokines in Asthma

Neutrophils also secrete cytokines and chemokines, such as IL-1β, TNF-α, CXCL8 (IL-8), and GM-CSF, which further enhance the inflammatory response leading to chronic inflammation, airway hyperresponsiveness and, in a longer perspective, airway remodeling (Figure 2) [89,108,109,110,111].

#### 4.2.1. TSLP

Thymic stromal lymphopoietin (TSLP) is an alarmin produced by epithelial cells in response to irritants (e.g., allergens, viruses, bacteria, pollutants, and smoke) [110]. Compared to healthy controls, TSLP expression is increased in the airways of patients with asthma [111,112]. It mainly drives the Th2 type of inflammatory response, by inducing dendritic cells to express OX40 ligand (OX40L) which is required for triggering naive CD4(+) T cells to produce IL-4, IL-5, and IL-13 [113,114]. In addition, CD8+ cytotoxic T lymphocytes also produce IL-5, IL-13, and IFN-γ as a result of additional stimulation by CD40L [115]. Furthermore, TSLP also contributes to the development of non-eosinophilic inflammation. It is suggested that human TSLP and TLR3 ligands may promote Th17 cell differentiation through activation of dendritic cells [116]. It has also been documented that bronchial epithelial cells increase neutrophil recruitment through IL-8 and GM-CSF production [117].

#### 4.2.2. IL-17

Interleukin 17, a key cytokine in neutrophilic asthma, is produced by Th17 cells. It elevates expression of cytokines and factors which propagate neutrophil recruitment and activation into the airways via IL-6, IL-8, and TNF- α [118]. Furthermore, it induces bronchial remodeling by stimulating fibroblasts to produce collagen and growth factors, such as transforming growth factor-beta (TGF-β). IL-17 is overexpressed in bronchial mucosa of patients with neutrophilic asthma, particularly in cases with more pronounced neutrophilic inflammation and frequent exacerbations [11]. In another study, patients with severe asthma present increased levels of IL-17 in bronchial biopsy as well as in sputum [119,120]. A correlation has been demonstrated between IL-17 mRNA levels and IL-8 mRNA levels, as well as neutrophil counts, indicating an association between IL-17 and neutrophilic inflammation [120]. IL-17 may be associated with glucocorticoid resistance as it induces glucocorticoid receptor (GR) β in epithelial cells in asthma [121].

#### 4.2.3. IL-8

Interleukin 8 (CXCL8/IL-8), secreted by bronchial epithelial cells in response to the pro-inflammatory cytokines IL-17, TNF-α, and IFN-γ, functions as a chemoattractant of neutrophil accumulation, promoting their survival in the airways [122,123]. Several studies have confirmed that IL-8 levels are significantly higher in patients with asthma than in healthy subjects [124,125,126]. In asthma, levels of IL-8 are correlated with the percentage of neutrophils in BAL, which points to the role of IL-8 in the recruitment and activation of neutrophils in the airways [127]. The highest IL-8 levels have been found to be associated with the non-eosinophilic asthma phenotype [16]. Furthermore, a significant increase in IL-8 levels was observed in induced sputum children during an asthma exacerbation, in comparison to the stable period of the disease. This finding is in line with what is known about the role of neutrophil recruitment during airway infections [128].

#### 4.2.4. IL-1β

Interleukin-1β (IL-1β) is a potent pro-inflammatory cytokine which is produced and secreted by a variety of cell types, particularly by innate immune system cells, such as monocytes and macrophages. It is produced as an inactive precursor, termed pro-IL-1β, in response to PAMPs [129]. Consecutively, pro-IL-1β is cleaved by the pro-inflammatory protease caspase-1 [130]. *Chlamydia* spp. and *Haemophilus* spp. infections have been shown to induce responses with increased IL-1β, NLRP3, and caspase-1, that drive steroid-resistant neutrophilic inflammation and airway hyperresponsiveness.

It has been demonstrated that IL-1β promotes Th17 differentiation and IL-17 production. Therefore, an increased level of IL-1β is a marker of more severe asthma, as it induces steroid-resistant neutrophilic inflammation and airway hyperreactivity [131,132]. This finding was also confirmed in a mouse model of asthma. Neutrophilic airway inflammation, disease severity, and steroid resistance have been shown to correlate with NLRP3 and IL-1β expression. Furthermore, treatment with anti-IL-1β antibodies resulted in both inhibition of the IL-1β response and remission of clinical characteristics of steroid-resistant asthma, while IL-1β administration led to the restoration of these characteristics [131].

#### 4.2.5. IL-33

IL-33 is passively released as an ‘alarmin’ upon tissue injury or necrosis, predominantly secreted by structural cells at barrier tissues [133]. Although it is a cytokine that mainly drives T2-high inflammation through the activation of immune cells such as mast cells, eosinophils, and ILC2, there are also studies demonstrating its role in non-eosinophilic asthma [134]. IL-33 acts through the ST2 receptor, which is present in conventional and regulatory T cells, ILC2, M2 macrophages, mast cells, eosinophils, basophils, neutrophils, NK cells, and iNKT cells [135]. Following IL-33 stimulation, peripheral neutrophils show increased MPO expression and phosphorylation of ERK and MAPK [136]. It has been reported that neutrophil inflammatory proteases produce a shorter form of IL-33, which has 10–30 times more potent effects than full-length IL-33 in activating target cells [137,138]. This suggests that IL-33 can be more efficient in neutrophilic inflammation.

It has been found that levels of IL-33 and ST2 receptor were elevated in groups characterized by uncontrolled asthma symptoms and low eosinophilia. In a mouse model, blockade of the IL-33/ST2 axis resulted in reduction in neutrophilic inflammation, IL-17A, MPO, and NET marker (SA100A9) and also reduced the severity of symptoms [136]. In another study on a mouse model of rhinovirus-induced chronic asthma, it was observed that early neutrophilic inflammation with NET formation was found to be IL-33-dependent. In nasal samples from rhinovirus-infected patients with asthma, IL-33 levels correlated with neutrophil elastase and dsDNA, which was not found in healthy controls [139].

### 4.3. Smooth Muscles in Neutrophilic Asthma

In asthma, pro-inflammatory environment affects the structure and functions of bronchial smooth muscles (BSM) cells. They are major effectors of exaggerated airway narrowing and hyperresponsiveness. Their contraction occurs in response to several stimuli, e.g., allergens, pollution, cold air, or physical exercise. Bronchial hyperresponsiveness has been described to be controlled by T2-related cytokines such as IL-5 and IL-13. In neutrophilic asthma, pro-inflammatory cytokines IL-1β, IL-17, and IL-23 may also further increase this process [140]. Moreover, BSM stimulated with them secrete several chemokines (CXCL1, CXCL10, CCL10), recruiting immune cells and prolonging inflammation.

The development of long-standing asthma is characterized by airway remodeling, which encompasses the hypertrophy and proliferation of smooth muscle cells, culminating in the thickening of the bronchial walls [141]. This process is promoted by neutrophil-derived matrix metalloprotease-9 (MMP-9) and elastase [142,143]. The levels of elastase inversely correlate with pulmonary function [144]. Moreover, Th17-related cytokines such as IL-17A, IL-17F, and IL-22 have been demonstrated to possess the capacity to induce BSM cell migration in a receptor-selective manner. Furthermore, the presence of Th17 cytokine receptors (IL-17RA, IL-17RC, and IL-22R1) in these cells has been confirmed. It is noteworthy that IL-17A and IL-17F exert their effects through signaling pathways other than the IL-22 pathway, suggesting the existence of more than one mechanism in the pathogenesis of remodeling [145]. Moreover, IL-23 prolongs the expression of the listed Th17 cytokines that induce tissue pathology and chronic inflammatory diseases, giving prominence to the IL-23/IL-17 immune axis [146].

One of the most significant contributors to airway remodeling is transforming growth factor (TGF-β). In bronchi, it is secreted by several types of cells, including neutrophils. TGF-β is profibrotic cytokine [147]. Although neutrophils constitutively express TGF-β, those derived from the asthma population secrete higher amounts of this cytokine [148].

## 5. Management of Neutrophilic Asthma—Present and Future

Current asthma treatment focuses on a personalized approach, integrating pharmacological and non-pharmacological strategies to optimize disease control. The latter include lifestyle modifications, allergen avoidance, and smoking cessation in order to improve long-term outcomes. According to the GINA 2025 guidelines, ICS remain the cornerstone of therapy, effectively reducing airway inflammation and preventing exacerbations [1]. For patients with moderate-to-severe asthma, combination therapies with addition of LABA, LAMA, and LTRA are recommended. The evolving understanding of asthma phenotypes allows for more tailored treatments, ensuring better disease management and quality of life for patients. Thus, in cases of severe treatment-resistant asthma, biologic therapies targeting key inflammatory pathways (e.g., IL-4, IL-5) are widely used. They have shown significant benefits in reducing exacerbation rates and steroid dependency [1]. However, the vast majority of biologicals used in today’s practice target only T2-high inflammation (anti-IL-5, anti-IL-5R, anti-IgE, etc.). The only biological drug registered for use in asthma irrespective of the inflammatory pathway involved is tezepelumab. In neutrophilic asthma it has been demonstrated that, by binding TSLP, tezepelumab inhibits the TH17/neutrophil cascade at the beginning of the immune pathway, limiting inflammation and preventing the airway remodeling typical for steroid-resistant neutrophilic asthma [149].

Increasing knowledge about the biology of neutrophils and their role in asthma results in new potential therapies that could ameliorate control of neutrophilic asthma in the near future. Below, we present some therapies targeting neutrophilic inflammation that have been investigated recently.

### 5.1. Anti-IL-17: Brodalumab

Brodalumab (AMG 827) is a human anti-IL-17RA immunoglobulin G_2_ (IgG_2_) monoclonal antibody which binds with high affinity to human IL-17RA. This blocks the biological activity of IL-17A, IL-17F, IL-17A/F heterodimer, and 17E (IL-25).

In the randomized, double-blind, placebo-controlled trial, 302 patients with inadequately controlled moderate-to-severe asthma were treated with brodalumab at doses of 140 mg, 210 mg, 280 mg, or placebo. Significant improvement in ACQ was only seen in the subgroup characterized by high obstruction reversibility (≥20% improvement in FEV_1_ after β- agonists), and treated with a dose of 210 mg. There was no effect in the 280 mg group [150]. Another study (NCT 01902290) was terminated earlier due to the lack of efficacy observed following interim analysis. Therefore, brodalumab was deemed ineffective, and it was not considered to have a therapeutic role in the treatment of asthma, including asthma with a neutrophilic phenotype [151].

### 5.2. Anti- IL-1β

The activity of IL-1β, which is released by alveolar macrophages, is significantly higher in patients with asthma than in healthy subjects [152]. IL-1ra is a natural anti-inflammatory cytokine that competes with IL-1β for binding to the type I IL-1 receptor [153]. It has been shown that IL-1ra inhibits neutrophilic inflammation induced by both LPS and IL-1 [154]. Anakinra is a recombinant form of human IL-1ra, the use of which in patients with asthma has been studied by Hernandez et al. [153]. Patients who received two subcutaneous doses of anakinra (1 mg/kg, maximum dose 100 mg) were subjected to an inhaled LPS challenge, and the number of neutrophils in their sputum was assessed. Compared with the placebo group, those receiving anakinra experienced a significant reduction in airway neutrophilia. Furthermore, LPS-induced IL-1β, IL-6, and IL-8 concentrations were reduced by 39%, 83%, and 150%, respectively, during the anakinra treatment period compared with placebo [153]. These data may be relevant for the treatment of non-T2 asthma.

There were several studies planned to evaluate anakinra as a rescue treatment for acute allergic reactions (NCT 03513471, NCT 04035109). However, they were abandoned during COVID-19 pandemic. An ongoing trial (NCT 04278404) aims at assessing the safety and pharmacokinetics of anakinra in children.

### 5.3. Targeting IL-8 (CXCL-8)

IL-8 is one of the main chemotactic factors induced by Th17, and it works via the chemokine receptor 2 (CXCR2), along with the leukotriene B_4_ [155]. CXCR2 is a key regulator of neutrophil migration and, therefore, has been the most studied target in preclinical studies.

In a randomized, double-blinded study, SCH 527123 resulted in a mean reduction of 36% in sputum neutrophil count in 22 patients with severe asthma and sputum total cell count < 10 × 10^6^/g and neutrophils > 40% compared with a 6.7% increase in the placebo group [156]. The mean absolute blood neutrophil count initially decreased but returned to normal at week 5. There were fewer mild exacerbations of asthma (1.3 versus 2.25, *p* = 0.05) and an improvement in the ACQ score (by an average of 0.42 points). No statistically significant changes in FEV_1_ were observed. The drug was deemed safe for use in patients with severe asthma [157].

Another CXCR2 antagonist that was tested was AZD8309. Twenty healthy subjects were given a placebo or AZD8309 (300 mg) orally twice daily for three days, after which they were given inhaled LPS and their induced sputum was collected. AZD8309 was found to result in an average 77% reduction in sputum total cell count and 79% reduction in neutrophils compared with placebo after the LPS challenge [156].

These studies led to a clinical trial of AZD5069, which is another selective CXCR2 receptor antagonist. Of the four groups of adult patients with uncontrolled asthma, three received AZD5069 at different doses, while one was given a placebo. While treatment with this agent was well tolerated, none of the AZD5069 doses reduced the incidence of severe exacerbations compared with the placebo. According to the authors, these results call into question the role of CXCR2-mediated neutrophil recruitment in exacerbations of severe refractory asthma [158].

### 5.4. Targeting IL-6

Several human and animal model studies have demonstrated pleiotropic activity of IL-6 in terms of airway inflammation. Overexpression of IL-6 was associated with a lower FEV_1_ in patients with mixed eosinophilic–neutrophilic bronchitis. On the contrary, the loss of IL-6 signaling abolished the increase in eosinophil- and neutrophil-recruiting cytokines and chemokines, as well as allergen-induced airway inflammation in mice. These findings imply that asthma exacerbations, particularly those involving eosinophil–neutrophil inflammation, may respond to therapies targeting the IL-6 pathway [159].

Tocilizumab is a human monoclonal antibody that blocks IL-6 signaling. A small proof-of-concept study investigated whether a single dose of tocilizumab could prevent bronchospasm induced by inhaled allergens in patients with mild asthma but found no evidence to support this. However, effective blockade of IL-6 was demonstrated by a decrease in levels of CRP, IL-6, and soluble IL-6R. The presence of a biological effect but a lack of clinical effects means that this type of therapy requires further and more thorough testing on different patient groups [160]. One such study is being conducted now in the USA and Canada (NCT 04278404) to assess the safety and pharmacokinetics of tocilizumab in pediatric asthma.

Another option of IL-6 targeting treatment includes clazakizumab being tested in adults and in teenagers with severe asthma (phase 2 PRECISE study) and FB704A being tested in adults with severe asthma (phase 2 study NCT 05018299). Results of these trials have not been published so far [161].

### 5.5. Biologics Targeting the IL-33 Pathway

Astegolimab, a human IgG_2_ antibody, selectively inhibits the IL-33 receptor, ST2. In a randomized clinical trial, astegolimab reduced annual asthma exacerbation rate in a broad population of patients, including those with eosinophil-low phenotype, with inadequately controlled, severe asthma [162]. The drug remains a promising treatment option, but no phase III trial has yet been registered, despite the publication of a phase IIb study in 2021.

Itepekimab, an IgG_4_P monoclonal antibody targeting IL-33, has been shown to improve asthma control and lung function when administered subcutaneously at a dose of 300 mg every two weeks [163]. Current clinical trials are mainly conducted for COPD (AERIFY-1 and AERIFY-2) [164]. Clinical trials in patients with atopic dermatitis have been terminated due to the drug’s insufficient efficacy, as determined by an unblinded evaluation of proof-of-concept data [165].

Torozakimab, another anti-IL-33 antibody, was assessed in FRONTIER-3 study. The primary endpoint was changed from baseline to week 16 in pre-bronchodilator FEV_1_ in the group of patients with moderate-to-severe asthma. This effect was not seen in a study group as a whole. However, it was confirmed in a subgroup of frequent exacerbators (at least two episodes in the previous 12 months) [166].

To conclude, there are some emerging therapies that can ameliorate prognosis in patients with severe NA in the near future. Surprisingly, the most promising options are drugs with targets that are not specific for neutrophilic inflammation (e.g., anti-TSLP). On the contrary, drugs targeting cytokines strongly associated with neutrophils’ function (IL-1, IL-17) have not been proven highly effective. This raises concerns about the inclusion criteria applied in clinical trials. Effectively, in most of them, researchers recruited a general asthmatic population characterized by moderate-to-severe symptoms, high exacerbation rates, and poor response to ICS irrespective of inflammatory phenotype. Thus, poor responses acquired in several studies should not discourage the conducting of clinical trials in well-defined cohorts with confirmed NA.

## 6. Unmet Goals and Directions of Future Studies

Despite the better understanding of neutrophilic inflammation regulation, there are still important unmet needs in the field of neutrophilic asthma.

First, in terms of diagnosis of NA, sputum or blood neutrophil counts do not seem sufficient. There is a need for reliable biomarkers characterized by both good performance (sensitivity, specificity) and wide availability. The ideal biomarker of NA should correlate with the intensity of neutrophilic inflammation so that it can facilitate the diagnosis of NA and objective monitoring of the clinical course of the disease. Potential candidates include cytokines related to function of neutrophils (IL-1, IL-17); chemokines (CCL-4, CXCL-8); substances secreted by neutrophils (MMP-9, MPO); chitinase-like protein YKL-40; serum amyloid A [167,168,169,170,171]. Expression of the aforementioned biomarkers is significantly higher in NA compared to other phenotypes. However, their correlation with clinical outcomes, predictive value in relation to exacerbations, or response to biologicals remain unclear.

Another aspect that requires further investigation is the clinical relevance of neutrophilia in patients with asthma. As neutrophilia is not specific and may stem from several acute conditions (infections), comorbidities (obesity, endocrinopathies, etc.), therapies (systemic steroids), or lifestyle (e.g., smoking), it should be interpreted cautiously. In the future, we will need longitudinal studies on the stability of neutrophilic inflammation in asthma and relations between neutrophilia, asthma, ICS therapy, and comorbidities. In our view, the proper defining of neutrophilic phenotype and phenotyping of patients in daily practice will only be possible if we better understand the divisions between ‘primary neutrophilic asthma’ and ‘asthma with concomitant neutrophilic pathology’.

Finally, better understanding of neutrophilic inflammation in asthma may result in the development of tailored therapies. Despite finding potential targets important for neutrophilic inflammation, so far no neutrophil-specific drug has been found to be highly effective. As a matter of fact, we lacked clinical trials with clearly defined subgroups with NA. The poor results of the studies mentioned in this paper may stem from the enrolment of diverse populations with severe asthma instead of patients with confirmed NA who would benefit the most from neutrophils-targeted interventions.

## 7. Conclusions

Neutrophilic asthma is related to higher exacerbation rate, lower level of symptoms control, and poorer response to steroids compared to eosinophilic phenotype. It is further complicated by comorbidities such as obesity or GERD that may contribute to the development and severe course of the disease. The most important directions for future studies include the formulation of clear and applicable diagnostic criteria, choosing biomarkers that facilitate diagnosis and monitoring, and development of drugs targeting neutrophilic inflammation.

## Figures and Tables

**Figure 1 jcm-14-07137-f001:**
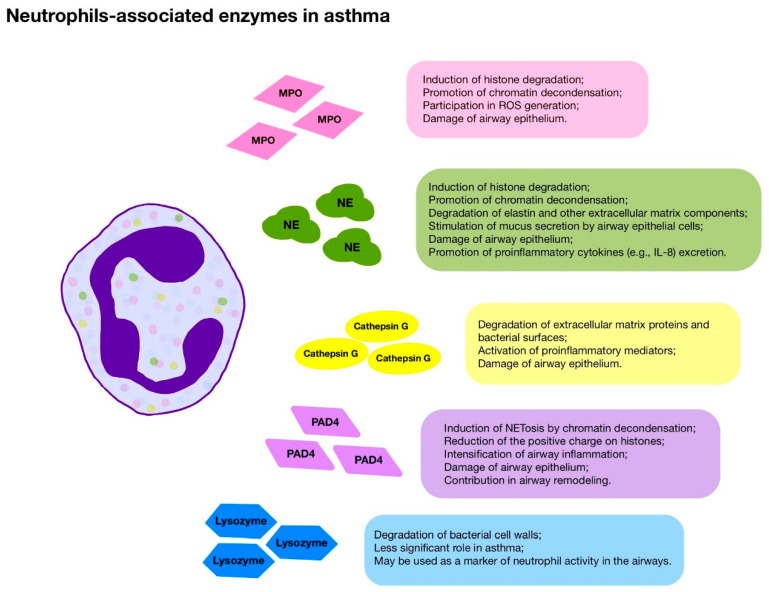
Neutrophils-associated enzymes in asthma.

**Figure 2 jcm-14-07137-f002:**
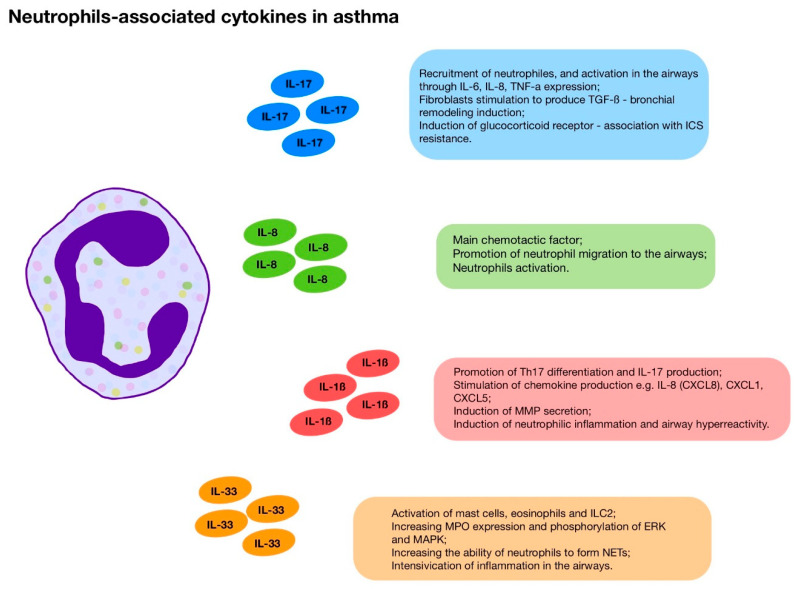
Neutrophils-associated cytokines in asthma.

**Table 1 jcm-14-07137-t001:** Cut-offs used by several authors to define the population with neutrophilic asthma.

Study	Population Included	Material Assessed	Neutrophilic Asthma Diagnostic Criteria
Green et al. [4]	259 patients with asthma 34 healthy controls	induced sputum	neutrophilia > 65%
Moore et al. [6]	423 severe asthma patients	induced sputum	neutrophilia > 40%
Simpson et al. [5]	93 patients with asthma 42 healthy controls	induced sputum	neutrophilia > 61%
Bullone et al. [11]	70 patients with mild-to-severe asthma	bronchial biopsies	47.17 neutrophils/mm^2^
Belda et al. [12]	118 healthy non-smokers	induced sputum	repeated neutrophilia > 65% or ≥5 × 10^9^/L (at least twice)
Demarche et al. [7]	833 patients with asthma 94 healthy subjects	induced sputum	neutrophil count ≥ 76%
Grunwell et al. [9]	68 children with severe asthma	BAL	neutrophil count ≥ 5%
Brooks et al. [13]	194 patients with asthma 243 non-asthmatics	induced sputum	neutrophilia 62–76% depending on age
Schleich et al. [14]	508 patients with asthma	induced sputum	neutrophil count ≥ 76%
Gibson et al. [15]	56 non-smoking patients with asthma	induced sputum	neutrophilia > 64%, 283 × 10^6^/mL
Shi et al. [16]	232 patients with asthma	induced sputum blood	neutrophilia > 61% blood neutrophil rate > 69%

**Table 2 jcm-14-07137-t002:** Comparison of major clinical characteristics of eosinophilic and neutrophilic asthma.

	Neutrophilic Asthma	Eosinophilic Asthma
Asthma onset	In adults, often after the age of 40	Mostly before the age of 40, very often in children and young adults
Asthma triggers	Infections, air pollution, smoking	Allergens
Immunological mechanism	T2-low	T2-high, often IgE-mediated
Response to ICS	Poor	Very good
Clinical course	Quite often severe, exacerbation-prone	In most patients easy to control with standard therapy
Biomarkers used in practice	None	Blood eosinophilia FeNO tIgE
Concomitant Diseases	Obesity Smoking GERD	Allergic rhinitis Atopic dermatitis Chronic sinusitis

## Data Availability

Data availability is not applicable to this article.

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
