# Peer review of "Neutrophilic Asthma—From Mechanisms to New Perspectives of Therapy"

_jcm, 2025, doi:10.3390/jcm14207137_

Round 1
Reviewer 1 Report
Comments and Suggestions for Authors
Dear Authors,
first of all, I thank you for giving me the opportunity to read this your manuscript.
I have only minor suggestions.
1) Some acronyms have been specified when used for the first time; many others did not. Most unspecified acronyms were not present in the List of Abbreviations. Please correct.
2) There are spacing errors (for example, see line 15 and line 73). Please correct.
3) The interferences of ICS therapy on the diagnosis of neutrophilic asthma - as suggested, for example, by the article by Cowan - need to be better clarified.
Author Response
Please, see the attachment.

Reviewer 2 Report
Comments and Suggestions for Authors
This manuscript is a well-organized and detailed review of neutrophilic asthma (NA). It provides depth on mechanisms and emerging therapeutic targets, which is valuable for clinicians and researchers. However, several areas require revision to ensure scientific rigor, person-first language, and sharper focus on the controversies around whether NA is a distinct phenotype.
Major Comments
Person-First Language
Throughout the text, the authors frequently use “asthmatics.” Person-first language (“patients with asthma”) is preferred in modern clinical writing for inclusivity and patient-centeredness. Please revise consistently.
Definition of Phenotype vs. Comorbidity Artifact
At line 241 and elsewhere: smoking, obesity, and GERD are repeatedly associated with airway neutrophilia. However, smokers in general (with or without asthma) show elevated sputum neutrophils. This raises the possibility that “neutrophilic asthma” may not be a true, stable phenotype, but rather a reflection of comorbidities, exposures, or treatments (e.g., corticosteroids). The manuscript should grapple more directly with this possibility instead of assuming NA is always a distinct endotype.
I suggest dedicating a section to “Is neutrophilic asthma a phenotype or a lab artifact?” and critically review evidence for/against.
Overemphasis on General Neutrophil Biology
While the background on neutrophil development and functions (phagocytosis, degranulation, NETosis) is accurate, much of it is textbook material and does not directly advance understanding of NA.
I suggest the authors condense general neutrophil biology and expand the sections specifically addressing neutrophil roles in airway inflammation, steroid resistance, and remodeling in asthma.
Interpretation of Treatment Trials
The depth of coverage of therapeutic targets is excellent. However, the narrative often emphasizes positive findings without equally highlighting negative or failed results. Explicitly contrast successes (e.g., macrolides, tezepelumab) with failures and analyze why neutrophil-targeted therapies are challenging.
Prevalence and Definitions
The wide variability in prevalence estimates (4–57%) is presented descriptively but not adequately explained. It would help to emphasize that cutoffs for “neutrophilia” differ (BAL vs. sputum vs. blood), that inflammatory profiles fluctuate over time, and that corticosteroid exposure shifts the phenotype. Would suggest to explore this variability more, and maybe synthesize the data and suggest a clinically relevant cutoff?
Future Directions
The conclusion should present a sharper research agenda. Som suggested priorities:
Development of robust biomarkers beyond sputum neutrophils.
Longitudinal studies on stability of neutrophilic inflammation.
Better accounting for comorbidities and lifestyle factors in defining phenotypes.
Stratification of trial populations to identify subgroups most likely to benefit from neutrophil-targeted therapy.
Minor Comments
- Ensure consistency when using “non-eosinophilic asthma” versus “neutrophilic asthma.” These terms are not interchangeable.
- Line 129: The statement about the Moore study needs a reference
- Consider trimming repetitive mechanistic details to reduce length.
- Some mechanistic statements (e.g., IL-33 effects on neutrophils) could be improved by citations from the most recent literature.
Reviewer 3 Report
Comments and Suggestions for Authors
- Too descriptive, too general, reads more like an extended introduction.
- Needs a clearer structure: background, problem, what the review adds, conclusion.
- Should emphasize novelty (what makes this review different from previous ones)
Introduction
- Needs a sharper focus: briefly summarize general asthma, then move quickly to neutrophilic asthma (NA).Reducing the general background of asthma (already well known) and expanding on why NA is clinically important.
- Missing a short paragraph on knowledge gaps (e.g., lack of biomarkers, poor response to ICS, unmet therapeutic need).
- In line 66, add the reference
- In line 97, the sentence is not clear
Definition and Prevalence of NA
- Good collection of studies, but overly detailed (especially the table with cut-offs).
- Needs critical appraisal: why such heterogeneity exists, and what are the implications for clinical practice.
- condense numerical details, emphasize variability and challenges in diagnosis.
Clinical Characteristics
- Well referenced but again too descriptive., For example, the section on obesity is too long
- Missing comparative summary with eosinophilic asthma (a concise table or paragraph would help).
- Needs more critical discussion of inconsistencies between studies (some show NA linked to exacerbations, others not).
- Could highlight clinical uncertainty and implications for treatment decisions.
Mechanisms of Neutrophilic Asthma
- Very detailed and again long, no need for basic knowledge about neutrophils ( lines 258-300 is not needed), also the “ classical role is too long. Figure 1 is not needed. The authors should focus on the roles of neutrophils in asthma instead
- In figure 2 there are 2 Titles: Activity of neutrophils-derived enzymes in asthma and Neutrophil-associated enzymes in asthma. Make one title
- 4.4. Resistance to ICS does not belong in the section Mechanism, it can be named differently such as Clinical Consequences and ., Why these mechanisms explain steroid resistance
- Missing a section on heterogeneity of neutrophils (different phenotypes, pro-inflammatory vs. pro-resolving).
Management and Therapies
Overloaded with descriptions of drugs and trials, more descriptive than critical
In lines 581 and 583 it is not clear why asthma exacerbations and bronchoconstriction are underlined
Line 673 Anty is spelled incorrectly
Missing critical analysis: which therapies are most promising, which failed, and why.
Add more recent references (2022–2024) for biologics and small molecules..
There is no Section about the unmet needs (lack of biomarkers, limited biologics for T2-low asthma), Limitations of existing studies (small cohorts, heterogeneity of definitions), Clinical challenges (overlap with COPD, variable neutrophilia), future perspectives (personalized therapy, emerging biomarkers, ongoing trials).
Conclusion
Too general and weak, too broad and too long. Should emphasize why NA matters for clinicians, the most urgent research directions and Clear take-home messages (1–2 sentences max).
References
Needs more recent sources (2022–2024), especially for new biologics and ongoing clinical trials.
Round 2
Reviewer 3 Report
Comments and Suggestions for Authors
Dear Authosr,
the new version is substantially improved: the introduction is more focused, the definition and prevalence section now includes a critical discussion, and the clinical part has been strengthened with a comparative table and comments on inconsistencies. The mechanistic section has been streamlined, redundant material removed, and neutrophil heterogeneity emphasized. The therapy section now contains updated references and a more balanced critical appraisal. Importantly, a new section on unmet needs and future perspectives has been added, and the conclusion has been rewritten to provide concise take-home messages.
Overall, the manuscript is now more focused, critical, and clinically relevant. I have no further major comments.
Author Response
Dear Reviewer,
Thank you for your remarks and suggestions on how to improve our manuscript. As we agree with all of them, we have introduced following changes:
- As requested, the whole section devoted to the role of neutrophils in immunity has been removed (page 8). Section 4 is now focused only on the role of neutrophils in asthma. Moreover, after implementation of this suggestion the manuscript is significantly shorter and easier to read.
- In the section of therapy we introduced several changes suggested by the reviewer. The list of possible therapies has been cut and now is limited to neutrophils-specific targets. We removed sections devoted to azithromycin and anti-TSLP therapies (pages 13-16).
- As you suggested, we have mentioned asthma phenotypes associated with concomitant diseases (‘obesity-related asthma’, ‘smokers’ asthma’ etc.) so that the readers are aware of the fact that patients with neutrophilic or paucigranulocytic asthma are often assigned to clinical instead of inflammatory phenotypes (page 8).